# Ferroptosis: At the Crossroad of Gemcitabine Resistance and Tumorigenesis in Pancreatic Cancer

**DOI:** 10.3390/ijms222010944

**Published:** 2021-10-10

**Authors:** Jianhui Yang, Jin Xu, Bo Zhang, Zhen Tan, Qingcai Meng, Jie Hua, Jiang Liu, Wei Wang, Si Shi, Xianjun Yu, Chen Liang

**Affiliations:** 1Department of Pancreatic Surgery, Fudan University Shanghai Cancer Center, Shanghai 200032, China; yangjianhui@fudanpci.org (J.Y.); xujin@fudanpci.org (J.X.); zhangbo@fudanpci.org (B.Z.); tanzhen@fudanpci.org (Z.T.); mengqingcai@fudanpci.org (Q.M.); huajie@fudanpci.org (J.H.); liujiang@fudanpci.org (J.L.); wangwei@fudanpci.org (W.W.); shisi@fudanpci.org (S.S.); 2Department of Oncology, Shanghai Medical College, Fudan University, Shanghai 200032, China; 3Shanghai Pancreatic Cancer Institute, Shanghai 200032, China; 4Pancreatic Cancer Institute, Fudan University, Shanghai 200032, China

**Keywords:** ferroptosis, gemcitabine resistance, NRF2, ROS, pancreatic ductal adenocarcinoma

## Abstract

The overall five-year survival rate of pancreatic cancer has hardly changed in the past few decades (less than 10%) because of resistance to all known therapies, including chemotherapeutic drugs. In the past few decades, gemcitabine has been at the forefront of treatment for pancreatic ductal adenocarcinoma, but more strategies to combat drug resistance need to be explored. One promising possibility is ferroptosis, a form of a nonapoptotic cell death that depends on intracellular iron and occurs through the accumulation of lipid reactive oxygen species, which are significant in drug resistance. In this article, we reviewed gemcitabine-resistance mechanisms; assessed the relationship among ferroptosis, tumorigenesis and gemcitabine resistance, and explored a new treatment method for pancreatic cancer.

## 1. Introduction

Pancreatic cancer is a digestive tumor with high mortality and metastasis rates that still rank among the top-10 list medical challenges [1,2,3,4]. By 2030, because of the general aging of the population and improvements in other cancer treatments pancreatic cancer is expected to surpass breast, prostate, and colorectal cancer as the second-leading cause of cancer-related death in Western countries after lung cancer [5].

Ferroptosis is a form of cell death that is different from apoptosis, necroptosis, or autophagy [6]. It is characterized by lipid peroxidation and depends on intracellular iron and the accumulation of reactive oxygen species (ROS) [7,8]. It has been found in many pathological diseases, such as ischemia-reperfusion injury, neurodegenerative diseases, and in different cancer types (e.g., hepatocellular, breast and pancreatic) [8,9,10]. It was discovered in a high-throughput screening campaign to identify molecules that can selectively induce cell death in isogenic cells carrying RAS mutant subtypes [11]. It was reported that Kirsten rat sarcoma virus oncogene (KRAS) mutations were present in approximately 95% of cases of pancreatic cancer and played an important role in its occurrence and development [12,13]. Therefore, there may be some special links between ferroptosis and the development of pancreatic cancer.

Pancreatic ductal adenocarcinoma (PDAC) is the primary pathological type of pancreatic cancer and has a poor prognosis [14]. Gemcitabine is the main chemotherapy treatment, but after a few weeks resistance begins to develop [15]; nevertheless, initial sensitivity of PDAC to gemcitabine provides patients with an opportunity for surgical treatment. Therefore, elucidating the molecular basis of gemcitabine resistance may help increase the effectiveness of this chemotherapy. At the same time, studies have shown that inhibiting cystine/glutamate antiporter (xCT) and glutathione peroxidase (GPX4)—key molecules related to ferroptosis—may be beneficial for inducing the clearance of cancer cells that are resistant to conventional chemo- or radiotherapy [16]. Therefore, it is particularly important to explore the relationship between ferroptosis and gemcitabine resistance not only for PDAC patients but also for those with, for example, biliary tract, breast or non-small cell lung cancer.

## 2. Entanglement between Pancreatic Cancer and Gemcitabine Resistance

### 2.1. Rapidly Developing Tumors and Drug Resistance Make Pancreatic Cancer Difficult to Treat

Surgery has so far increased the survival of pancreatic cancer patients. Unlike surgical methods in previous decades, which were highly risky and difficult, modern surgery in experienced centers is relatively safe [12,17] but is only suitable for 15–20% of cases. In addition, the post-operative tumor recurrence is high, and the local recurrence rate exceeds 50%. The randomized controlled trial showed that the 5-year survival rate of patients undergoing tumor resection is about 28%, and the median survival time is 18 months. The above data indicate that surgery as a treatment has limits. [15]. Therefore, in the past few decades, adjuvant treatment strategies have gradually been developed to improve survival of which chemotherapy is an indispensable part. Adjuvant chemotherapy after radical surgery can significantly improve the survival rate and reduce recurrence [18]. Solid tumors have high interstitial tension and poor blood perfusion during growth. This is especially apparent in pancreatic cancer, which have a low blood supply and show abnormal resistance to most chemotherapy drugs. Conventional intravenous chemotherapy often fails to deliver effective amounts of drugs to the tumor. If the total dose is too high, adverse reactions may weaken the patient’s immune system and reduce the drug’s therapeutic effect. For decades, researchers have made progress in diagnosis, surgery, chemotherapy, immunotherapy, but the overall prognosis of PDAC patients is still poor [12,19].

### 2.2. Variability of Drug Resistance in Pancreatic Cancer

Pancreatic cancer is still one of the most challenging research fields despite continuing efforts to find an effective treatment [20,21] for disease progression [22]. Treatment failure is due to a variety of factors, including resistance to traditional chemotherapy, which can be classified as extrinsic and intrinsic resistance [23,24]. Gemcitabine is usually the recommended first-line drug for PDAC patients, and can be combined with other drugs to enhance the efficacy of the chemotherapy [25]. External resistance from the surrounding dense tumor stroma is easier to understand because it acts as a biophysical rampart that segregates PDAC into tumor epithelial cells inside the stroma and keeps chemotherapy drugs like gemcitabine outside [26]. However, the mechanism of intrinsic resistance to gemcitabine is still an active area of research [27].

Gemcitabine, a nucleoside analog of deoxycytidine, was approved by the FDA in 1996 and is now widely used in the treatment of various solid tumors (breast, ovarian cancer and non-small cell lung cancers [28]) and is still the cornerstone of neoadjuvant, adjuvant and palliative treatment of PDAC [29]. The cytotoxic activity of gemcitabine (2′, 2′-difluoro 2′-deoxycytidine, dFdC) is based on several effects on DNA synthesis. Due to the structural differences between the fluorine substituents at the 2’ position of the furanose ring, gemcitabine has obvious therapeutic advantages over other nucleotide analogs in terms of cellular pharmacology, metabolism and mechanism of action [28]. It has a rapid deamination effect on its inactive metabolite, dFdU, and requires a series of phosphorylation steps to attain relevant activity. Therefore, gemcitabine has certain limitations as an alternative drug, especially due to its toxicity at high and repeated doses and the development of chemotherapy resistance [30].

Compared with other malignant tumors, the main histopathological feature of PDAC is the hyperplasia of the surrounding interstitium of the locally infiltrated tumor tissue, which distorts the normal structure of pancreatic tissue [31]. The pathological feature of PDAC is its dense fibroproliferative tumor matrix and the intertwined sparse and collapsed vasculature, which is conducive to the occurrence and development of drug-resistant phenotypes. The pancreatic cancer stroma is composed of cancer-associated fibroblasts (CAF), vasculature, extracellular matrix (ECM) and immune cells [32]. CAFs promote tumor development and drug resistance by regulating the microenvironment or directly regulating tumor epithelial cells [33]. Multiple studies have shown that the tumor stroma promotes tumor progression, invasion, metastasis and chemoresistance in PDAC [34]. Therefore, the PDAC matrix is also considered to be a factor affecting gemcitabine resistance. A large amount of evidence shows that the matrix affects the external resistance by impairing the delivery of gemcitabine [35]. At the same time, research on the matrix-mediated intrinsic resistance mechanism of gemcitabine has also been topical in recent years.

### 2.3. Exploration of Some Pathways of Gemcitabine Resistance Related to Ferroptosis

#### 2.3.1. PI3K/Akt Pathway and MAPK Pathway

Pancreatic studies have shown that excessive activation of the PI3K/Akt pathway can lead to gemcitabine resistance [36]. This event was verified by taking several selective PI3K or Akt inhibitors (Figure 1). At the same time, these factors led to the recovery of Bax pool and gemcitabine-induced apoptosis [36,37,38]. Studies have shown that thymoquinone, an inhibitor of the Akt/mTOR/S6 pathway, not only enhances the sensitivity of oxaliplatin and cisplatin in pancreatic cancer but also similarly promotes sensitivity to gemcitabine [39]. Phosphatase and tensin homologue (PTEN), deleted on chromosome 10, is best characterized as an antagonist of the PI3K/Akt signaling pathway [40,41]. There is increasing evidence that upregulating PTEN helps increase the chemosensitivity of pancreatic cancer cells resistant to gemcitabine [42,43]. The Src tyrosine kinase pathway is another driver of the Akt pathway, and its role in PDAC is over-amplified by KRAS. Src/ErbB2 induces Akt activation and gemcitabine resistance in pancreatic cancer through amplification cycles [44,45], and apoptosis induced by the caspase signaling pathway is related to the MAPK pathway. The cytotoxic effect of gemcitabine may be mediated by the activation of p38, leading to MAPK–caspase-dependent apoptosis [46]. In recent years, studies have also found that ferroptosis is closely related to the MAPK pathway. The activation of STAT3 induced by ferroptosis inducer erastin in PDAC cells requires activation of the MAPK/ERK pathway [47]. Studies have also found that the ASK1–p38 MAPK pathway is one of the regulators of ferroptosis downstream of lipid peroxides [48]. Therefore, ferroptosis and gemcitabine resistance may also have a certain relationship in the PI3K/Akt and MAPK pathways.

#### 2.3.2. NF-kB and NRF2 Pathway

Nuclear factor-k-gene binding (NF-kB) was up-regulated after gemcitabine treatment in colon cancer, breast cancer and PDAC [49,50,51]. Data showed that approximately 75% of cases showed a loss of p53 function, and gemcitabine upregulated the level of NF-kB in a dose-dependent manner. Accordingly, inhibition of the NF-kB pathway in PDAC cells diminished gemcitabine resistance to varying degrees [52,53,54]. The research findings provided a combined strategy for regulating cell redox status and overcoming gemcitabine resistance, resulting in a synergistic and selective cytotoxic effect in vivo. Recent reports indicate that NF-kB and Nuclear factor erythroid-2-related factor 2 (NRF2) play an important role in the development of pancreatic cancer and chemoresistance [55]. The report described that gemcitabine activates NOX-induced ROS accumulation through NF-kB, and the increased ROS level can induce the activation of NRF2 and the increase of intracellular GSH, which leads to the intrinsic resistance of PDAC to gemcitabine. Gemcitabine stimulates the generation of endogenous ROS, and phenethyl isothiocyanate (PEITC) inhibits the elimination of ROS. The combination of the two induces strong oxidative stress, which in turn induces the death of pancreatic cancer cells [27].

NRF2 is a transcription factor that can regulate the expression of genes involved in cell protection and anti-oxidation, thereby protecting cells from electrophilic or oxidative stress [56]. In the cell, NRF2 can bind to the Kelch-like ECH-related protein 1 (Keap1) in the cytoplasm and be degraded by the E3 ubiquitin proteasome. Recent studies have also shown that NRF2 overexpression is involved in the cell proliferation and chemoresistance of various cancers [57,58,59,60]. The deficiency of HEAT repeat containing 1(HEATR1) can promote pancreatic cancer proliferation and gemcitabine resistance by up-regulating NRF2 signal. Digoxin reverses gemcitabine resistance by inhibiting the NRF2 signal in SW1990/GEM and Panc-1/GEM cells [61,62]. Some researchers have used experiments at the level of related genes to prove that after PDAC cells are treated with gemcitabine additional NRF2 depletion reduces GSH levels and increases ROS production. In addition, knockdown of NRF2 with NRF2 siRNA found that cell lines with different PDACs were more sensitive to gemcitabine [27]. In addition, some researchers used PIK-75 to inhibit NRF2 and changed the sensitivity of pancreatic cancer cells to gemcitabine [63]. Studies also showed that reducing the glutathione pool to promote the redox regulation mechanism induced by gemcitabine may affect tumor cell survival and drug sensitivity. In addition, it was found that PEITC can consume GSH and subsequently strengthen the oxidative stress response [64,65], which may affect this pathway to enhance the sensitivity of gemcitabine to pancreatic cancer. Hence, NRF2 has a great clinical application value in gemcitabine resistance.

#### 2.3.3. HSP Pathway

Heat shock proteins (HSP) are a group of highly conserved proteins that belong to the class of molecular chaperones and are induced by different kinds of stress [66]. It has strong cell protection properties and can regulate the response of cells to various stresses. Some HSPs can produce cell protection against chemotherapy and other attacks, and some can drive gemcitabine-induced apoptosis. Heat shock transcription factor (HSF) can be regulated by HSP90, HSP70 and HSP27. Compared with normal tissues, it is overexpressed in several malignant tumors [29]. The data suggest that HSP90 inhibition is a highly effective therapeutic strategy in pancreatic cancer supported by its ability to inhibit antiapoptotic and proliferative pathways by synchronously interrupting multiple key oncogenic signaling cascades in gemcitabine and 5-FU-resistant pancreatic cancer [67]. Overexpression of HSP70 in sarcoma cells induces gemcitabine resistance, and modulation of HSP70 expression with quercetin (an HSF inhibitor) enhances the chemo-responsiveness of pancreatic cancer cells to gemcitabine [68,69]. In addition, when HSP27 was knocked down in resistant pancreatic cells (KLM1-R), sensitivity to gemcitabine was restored. In addition, the increased expression of HSP27 in tumor tissues was related to gemcitabine resistance in patients with pancreatic cancer [70]. Interestingly, when studying the high-throughput liquid chromatography–tandem mass spectrometry (LC/MS) assay to detect the drug resistance pattern in PDAC cells, it was found that HSP27 and nucleophosmin are closely related to gemcitabine resistance [71,72]. These results may suggest that HSPs play an important role in resistance to gemcitabine.

#### 2.3.4. miRNA-Related Pathway

MicroRNAs (miRNAs) are short non-coding RNAs with a length of 18–22 nucleotides, which are related to cell proliferation, cell cycle control, cell differentiation, migration, invasion, and chemotherapy resistance [73,74]. According to research, miRNAs are thought to have a regulatory role in pancreatic cancer resistance to gemcitabine and regulate related pathways such as KRAS, PI3K-AKT, NF-kB, P53, and Hedgehog [75]. MiR-21 expression is directly correlated with chemotherapy resistance and promotes gemcitabine resistance by targeting PTEN or by overexpressing matrix metalloproteinase 2 (MMP2), matrix metalloproteinase 9 (MMP9) and vascular endothelial growth factor (VEGF), which in turn induces the PI3K/AKT pathway. MiR-301a-3p confers resistance to gemcitabine by regulating the expression of PTEN [76]. Some miRNAs have also been found to be involved in the inhibition of gemcitabine-induced apoptosis. Pretreatment of CAFs with miR-106b inhibited their expression in CAF exosomes, thereby reducing gemcitabine resistance. In-depth research found that miR-106b promotes gemcitabine resistance of cancer cells by directly targeting TP53INP1 [77]. In addition, exosomal miR-222-3p can regulate gemcitabine resistance and maintain the malignant characteristics of pancreatic cancer by targeting SOCS3 [78]. Studies also found that the overexpression of miR-365 in aggressive PDAC can directly lead to gemcitabine resistance (Figure 1). It may affect the adaptor protein Src homology 2 domain contains 1 (SHC1) and the apoptosis-promoting gene Bax [79]. In addition, the overexpression of miR-200 that was concomitant with gemcitabine treatment reversed resistance [80]. MiR-506 enhanced apoptosis and chemosensitivity of pancreatic cancer cells by SPHK1/Akt/NF-kB signaling [81].

## 3. Ferroptosis Promotes Tumorigenesis in Pancreatic Cancer

Ferroptosis is a regulated form of cell death, mainly caused by the loss of activity of the lipid repair enzyme glutathione peroxidase 4 (GPX4) and the accumulation of lipid-based ROS, especially lipid hydroperoxide [6]. Studies have shown that the pathways involved in ferroptosis include iron metabolism and ROS metabolism pathways [16,82]. Morphologically, it was found that during ferroptosis, the volume of mitochondria decreased; the density of mitochondrial membrane increased; the mitochondrial ridges disappeared; and the outer membrane ruptured [83].

Iron metabolism in cells includes iron input, storage and output. After binding to transferrin, ferric iron (Fe^3+^) is transported to the endosome through transferrin receptor 1, and then is reduced to ferrous iron (Fe^2+^), and finally accumulates in the unstable iron pool in the cytoplasm. Cytoplasmic iron is mainly ferritin (Ft): heavy chain (FtH) and light chain (FtL). Finally, the excess iron is exported by an iron transporter. Compared with drug-sensitive cells, the uptake, storage and output of iron in drug-resistant cancer cells have changed a little [84,85]. In research about the relationship between iron metabolism and tumors, it was found that cancer cells have a large demand for iron, and iron promotes the formation and metastasis of tumor cells [86,87,88].

ROS belong to partially reduced oxygen-containing molecules, including superoxide (O_2_•–), peroxide (H_2_O_2_ and ROOH) and free radicals (HO• and RO•). Iron participates in the Fenton reaction in which Fe^2+^ reacts with hydrogen peroxide [85] generating hydroxyl radicals, another ROS. The mitochondria in the cell generates a large amount of ROS through normal metabolism in the electron transport chain and energy production [89]. ROS from ferroptosis can reduce the stability of DNA thereby promoting cancer cell death.

In the high-throughput screening of small molecule libraries, it was found that erastin and RAS selective lethal small molecule 3 (RSL3) can be used as ferroptosis-inducing compounds and cause oncogenic RAS mutant cells to be selectively lethal without inducing apoptosis [11,90]. Several molecules that regulate iron metabolism and lipid peroxidation have recently been discovered. Among them, system Xc- and GPX4 are negative regulators of ferroptosis [91]. After the cells are treated with erastin, the input of radiolabeled cystine (a substrate of the Xc-antiporter system) can be prevented, thus confirming that erastin inhibits the Xc- system from activating ferroptosis [8]. In addition, knocking down GPX4 can generate more lipid ROS and induce ferroptosis. This further proves that RSL3 induces ferroptosis by inhibiting GPX4 [92], which is not affected by oxidized polyunsaturated fatty acids (PUFAs) or fatty-acid free radicals. The overall survival analysis found that the high expression of GPX4 was associated with the increased survival of PDAC patients. All these indicated that GPX4 may be the main prognostic marker of PDAC [93]. Studies also found that the Xc-SLC7A11 cystine–glutamate antiporter system affects the intracellular level of the cofactor glutathione (GSH) of GPX4. It was also found that drugs such as sulfasalazine, sorafenib or glutamate can inhibit related transporters to induce ferroptosis [94].

In recent years, researchers have invented the engineered mouse model (GEMM) of PDAC [95]. Compared with the normal model, GEEM has KRAS mutations or other changes in tumor suppressor genes (such as *p53* or *Cdkn2a*) in mice. Although they have their own characteristics, there is no clinical pathology that can fully mimic PDAC. GEMM mainly includes two basic models: one is *Pdx1-Cre*; *Kras*^G12D/+^ mice (called KC), and the other is *Pdx1-Cre*; *Kras*^G12D/+^; T*p53*^R172H/+^ mice (called KPC). This model is widely used to study the signal, mechanism and treatment strategy of PDAC. Studies have found that the histopathological progress of KPC is faster than that of KC, especially for poor vascular distribution, fibrosis, local infiltration and metastatic dissemination [83]. The study noted that in KC mice with extra GPX4 depletion or a high-iron diet, the application of the iron death inhibitor lipoxstatin-1 can prevent the death of KRAS-driven mice and change the pathological and molecular changes that the pancreas prevents [93]. The depletion of pancreas SLC7A11 in KPC mice (compared to KC mice, which lacks mutant TP53) will produce different phenotypes, and the induction of iron death will prevent mutant KRAS/Tp53-induced pancreatic tumorigenesis [96]. Experimental studies have clarified the relevant mechanism. Oxidized nucleobases released from a high-iron diet or GPX4 consumption can induce the activation of transmembrane protein 173 (TMEM173, also known as STING) which is related to DNA sensor pathways. It can promote infiltration and activation of macrophages to cause the occurrence and development of pancreatic cancer [93] (Figure 2). GPX4 depletion or high-iron diet can induce iron death, and also increase the production and release of oxidized nucleobases (such as 8-OHG), which in turn promotes the accumulation and activation of macrophages, and produces abnormal cytokines, especially IL-6 and NOS2. Both are important participants in the development stage of PDAC and are associated with the low survival rate of pancreatic cancer [97]. A new discovery also indicates that zalcitabine, an antiviral drug for human immunodeficiency virus infection, can suppress the growth of primary and immortalized human pancreatic cancer cells through the induction of ferroptosis [98,99]. It relies on zalcitabine-induced mitochondrial DNA stress, which activates the STING1/TMEM173-mediated DNA sensing pathway. Moreover, researchers found that it promotes ferroptosis in human pancreatic cancer cell lines by increasing MFN1/2-dependent mitochondrial fusion [99]. At the same time, the study showed that ferroptosis could be triggered by inhibiting cystine import, GSH synthesis, or GPX4 in synergy with GOT1(cytosolic aspartate aminotransaminase) [100]. These verify the significance of ferroptosis in pancreatic cancer.

## 4. Induction of Ferroptosis Can Inhibit Gemcitabine Resistance

Ferroptosis has been found to play a role in different cancers, including head and neck, and pancreatic cancer and hepatocellular carcinoma. Metallothionein (MT)-1G is a key regulator of sorafenib resistance in human hepatocellular carcinoma (HCC) cells. The knockout of MT-1G by RNA interference technology will increase glutathione consumption and lipid overexpression. Oxidation, which induces ferroptosis and affects the drug sensitivity of sorafenib, has a certain effect on preventing the development of the disease [101]. These experiments proved the new molecular mechanism of sorafenib resistance and indicated that MT-1G may be a new regulator of ferroptosis in HCC cells [102,103,104]. Studies also showed that inducing ferroptosis can overcome cisplatin resistance in head and neck cancer (HNC). These findings indicate that ferroptosis in HNC cells involves cell death mechanisms related to cystine, glutamine, iron, glutathione and ROS [105,106]. Therefore, ferroptosis may also play a role in gemcitabine resistance in pancreatic cancer cells (Figure 3).

### 4.1. ROS at the Crossroad of Ferroptosis and Gemcitabine Resistance

Gemcitabine induces the accumulation of ROS during treatment, which is a newly discovered cytotoxic mechanism aimed at revealing the mechanism of its intrinsic drug resistance, which has also been a prominent topic of research in recent years [27]. Gemcitabine can induce the production of ROS, which is an additional anti-cancer mechanism [54,107]. Excessive production of ROS can cause cell damage and eventually cell death. Therefore, in the process of continuous cell evolution, the cell develops a highly self-regulated antioxidant defense system to combat oxidative damage. The antioxidant system for ROS in the cell is composed of various antioxidant enzymes, including the enzyme that produces glutathione (GSH), glutamate cysteine ligase (GCL), and glutathione reductase (GSR) And glutathione S-transferase (GST). [108,109]. Most of these enzymes are regulated by the transcription factor NRF2 [110]. Studies have found that gemcitabine induces GSH synthesis by activating NRF2 to promote ROS detoxification, and consumption of NRF2 can enhance the sensitivity of PDAC cells to gemcitabine (Figure 3). An in-depth study of related mechanisms found that gemcitabine treatment induced NF-kB activation and NOX-derived ROS accumulation through p22-phox expression in PDAC cells [111]. As a feedback mechanism, nuclear translocation of NRF2 stimulated the transcription of protective antioxidant genes, mainly those that encode enzymes to catalyze the production of glutathione (GSH) to reduce ROS levels, thereby resisting gemcitabine treatment. For further verification, the use of RNA interference (RNAi)-mediated NRF2 consumption or β-phenethyl isothiocyanate inhibits the detoxification process of ROS by reducing GSH levels, thereby improving the efficacy of gemcitabine in vitro and in vivo [112]. Therefore, when gemcitabine is used for treatment, it induces the production of ROS and may further activate the oxygen pathway to induce ferroptosis.

### 4.2. A Key Molecule between Ferroptosis and Gemcitabine Resistance

NRF2 is a major player in the regulation of antioxidant molecules in cells [113]. Indeed, it is an oncogenic transcription factor that plays a very important role in combating environmental or intracellular pressure and is responsible for regulating the cellular antioxidant system that produces GSH in cancer cells [114]. The data shows that most cancer cells show overexpression of NRF2, which is related to the poor effect of anti-cancer therapy and the low survival cycle of cancer patients. Interestingly, studies have found that the level of NRF2 in cells is closely related to the sensitivity of ferroptosis. Increased expression of NRF2 can prevent it, while decreased expression of NRF2 can enhance the sensitivity of cancer cells to iron death inducers [102,115,116]. Baicalein is a molecule found in some traditional Chinese herbal medicines and has powerful anti-cancer activity [117]. Baicalein showed higher anti-cancer activity in the ferroptosis of pancreatic cancer cells induced by erastin [118]. NRF2 positively regulates the key protein of ferroptosis, and can promote the transcription of its downstream targets, such as SLC7A11, G6PD and FTH176 [119]. These genes are involved in lipid peroxidation and iron metabolism and negatively regulate ferroptosis after transcription activation. Studies have shown that inhibiting the p62-Keap1-NRF2 pathway can significantly enhance the anti-cancer activity of erastin and sorafenib in HCC cells [102]. The research also demonstrated that plasma-treated, water-derived oxidants sensitize pancreatic cancer cells to ferroptotic cell death by targeting a NRF2–HMOX1–GPX4 specific kinase signaling network [120]. The above shows that NRF2 inhibitors and their downstream targets may be viable targets for inducing ferroptosis-dependent cancer cell death; in addition, they may also enhance the sensitivity of PDAC to gemcitabine. NRF2 inhibitors combined with ferroptosis inducers may be a feasible strategy for killing gemcitabine-resistant cells.

### 4.3. HSPA5 Regulates Ferroptosis to Inhibit Gemcitabine Resistance in Pancreatic Cancer

According to the molecular weight, HSPs can be divided into six families: HSP100, HSP90, HSP70, HSP60, HSP40 and small HSPs. HSPβ-1 is a negative regulator of ferroptosis, and its inhibited expression and phosphorylation enhanced erastin-induced ferroptosis in human xenograft mouse tumor models [121,122].

Heat Shock Protein Family A (Hsp70) Member 5 (HSPA5, also termed GRP78 or BIP) is a molecular chaperone protein mainly expressed in the endoplasmic reticulum, with a size of approximately 70 kDa [123]. As an important part of the unfolded protein response, HSPA5 can promote cell survival under endoplasmic reticulum stress conditions [124]. Recent reports clarified that upregulation of HSPA5 is a negative regulator of PDAC cell ferroptosis, prevents the degradation of GPX4, and inhibits lipid peroxidation during ferroptosis. Due to the low overall survival rate of PDAC, there are few other drug treatments for patients who have failed with gemcitabine. Studies found that when it is used in combination with HSPA5 inhibitors to treat PDAC, its anti-cancer activity is significantly increased, which is achieved by inducing ferroptosis [125]. At the same time, the increase in HSPA5 expression is closely related to the poor prognosis of PDAC patients [126]. Activating transcription factor 4 (ATF4) resulted in the induction of HSPA5, which in turn bound glutathione peroxidase 4 (GPX4) and protected against GPX4 protein degradation and subsequent lipid peroxidation. From this mechanism, this study showed that the upregulation of the HSPA5–GPX4 pathway contributed to gemcitabine resistance (Figure 3). In contrast, the use of RNAi, EGCG, or sulfasalazine to inhibit the genetic and pharmacological properties of the HSPA5–GPX4 pathway enhanced gemcitabine sensitivity in PDAC cells in vitro and mouse pancreatic cancer animal models [125]. These findings not only indicate the new role for HSPA5 in tumor ferroptosis, but also proposed a potential therapeutic strategy to overcome gemcitabine resistance in PDAC cells by inducing ferroptosis.

### 4.4. FBXW7 Potentiates Cytotoxic Effect of Gemcitabine via Ferroptosis

It has been reported that the use of erastin to induce the iron death pathway can enhance the cytotoxic effects of gemcitabine and cisplatin on pancreatic cancer cells [127]. The study also confirmed that the ferroptosis inducer RSL3 also increased the cell killing effect of gemcitabine in pancreatic cancer cells, while the ferroptosis inhibitor Fer-1 had the opposite effect. These experiments confirmed that the induction of ferroptosis can enhance the sensitivity of gemcitabine to pancreatic cancer cells and reduce its drug resistance. Studies tested an enzyme called FBXW7 (F-box and WD repeat domain-containing 7) to affect the sensitivity of gemcitabine [128]. FBXW7 can promote ferroptosis and cell apoptosis, and it is speculated that it may affect the cytotoxicity of gemcitabine through iron death and cell apoptosis. They used the iron death inhibitor Fer-1 and the apoptosis inhibitor zVAD to treat FBXW7 overexpressing cells, and found that the two treatments separately or in combination counteracted the effect of FBXW7 overexpression on gemcitabine. Therefore, it was confirmed that FBXW7 enhanced the cytotoxic effect of gemcitabine through ferroptosis and apoptosis (Figure 3), and that this effect was achieved through the FBXW7–NR4A1–SCD1 pathway [129]. Moreover, short-hairpin RNA-based knockdown of SCD1 enhanced erastin-induced ferroptosis in vitro under H/NS (Hypoxia and nutrient starvation) [130]. Furthermore, this provided direction for new treatment methods. It also pointed out the way for future research; that is, the drug resistance of gemcitabine can be studied through the ferroptosis pathway [128].

## 5. Conclusions and Perspectives

Ferroptosis has gradually become one of the key research topics in recent years and has important biological significance in the treatment of related diseases. Pancreatic cancer is one of the tumors with a higher mortality rate, and drug chemotherapy is an important treatment strategy. Data in the literature show that for pancreatic cancer, induction of ferroptosis combined with gemcitabine therapy had a very good effect.

Inhibiting key molecules in the ferroptosis pathway, such as cystine/glutamate antiporter (xCT) and glutathione peroxidase (GPX4), can eliminate cancers that are resistant to conventional chemo- or radiotherapy. For example, multiple experiments confirmed that inhibiting the HSPA5–GPX4 pathway will enhance the sensitivity of gemcitabine in PDAC cells [125]. It was also demonstrated that the use of drugs to inhibit the system x_c_- can enhance the cytotoxic effects of gemcitabine and cisplatin in PDAC cell lines [127]. Kenneth P. Olive et al. also confirmed that, after using KPC mice to induce ferroptosis, pancreatic cancer cells needed exogenous cystine to avoid ferroptosis and the loss of the previous SLC7A11 subunit to prolonging survival [96]. At the same time, because gemcitabine activates NOX-induced ROS accumulation through NF-kB activation, the increased ROS level led to the activation of NRF2 and the increase of cellular GSH, which led to the inherent resistance of PDAC. NRF2 inhibition combined with ferroptosis induction may be effective against gemcitabine resistance in PDAC. At the same time, many new technologies have been applied to treat tumors. For example, some researchers applied an intriguing nanomedicine strategy to achieve autophagy-enhanced ferroptosis to combat cancer [131].

Therefore, combining the above methods to induce ferroptosis in PDAC cells may become a research direction to reverse the resistance to gemcitabine and may provide a reasonable basis for the development of new therapies for pancreatic cancer. It is believed that in the near future, effective treatment with a combination of drugs will become a reality, and researchers will develop more strategies to combat cancer.

## Figures and Tables

**Figure 1 ijms-22-10944-f001:**
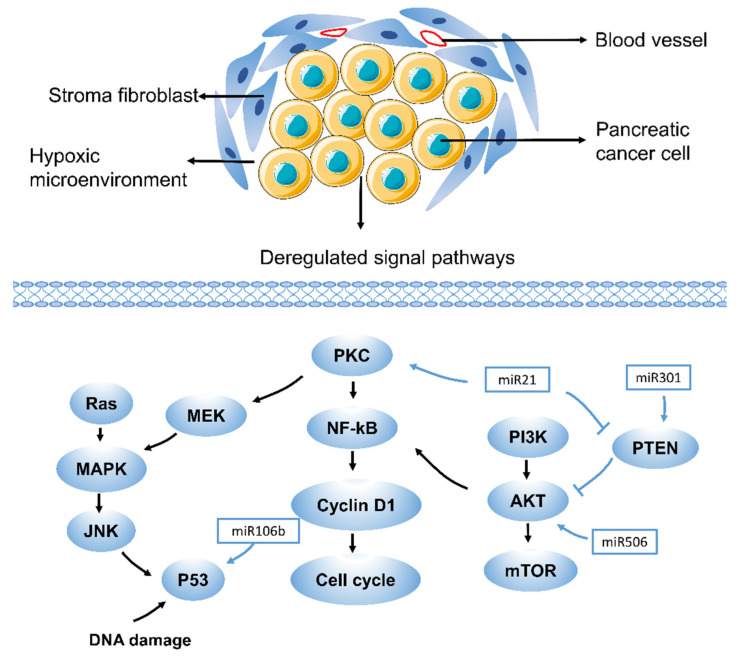
Drug resistance pathways in pancreatic cancer. RAS, rat sarcoma virus oncogene; JNK, Jun N-terminal kinase; NF-kB, nuclear factor-k-gene binding; PKC, Protein kinase C; PTEN, phosphatase and tensin homologue; PI3K, phosphatidylinositol 3-kinase. Drug resistance in pancreatic cancer is caused by various mechanisms, including aberrant gene expression, mutations, and deregulation of key signaling pathways (such as MAPK, Akt, NF-kB, and miRNA-related pathways). Each of these pathways contributes to drug resistance in pancreatic cancer in different ways, which suggests that different therapeutic targets exist. A few representative drug resistance pathways are shown, such as the MAPK, Akt, NF-kB, and miRNA-related pathways.

**Figure 2 ijms-22-10944-f002:**
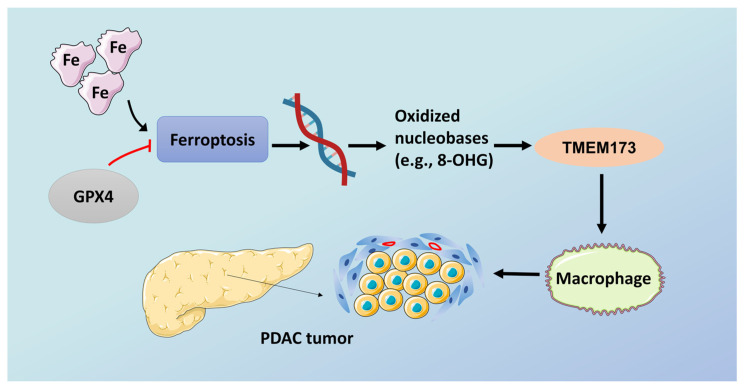
Ferroptosis promotes tumorigenesis in pancreatic cancer. GPX4, glutathione peroxidase 4; TMEM173, transmembrane protein 173. Schematic depicting the role of high-iron diets or GPX4 depletion in Kras-driven PDAC. The induction of ferroptosis by either high-iron diets or GPX4 depletion promotes oxidized nucleobase (e.g., 8-OHG) release and thus activates the TMEM173-dependent DNA sensor pathway, which finally results in macrophage infiltration and activation during Kras-driven PDAC. Consequently, macrophage depletion or pharmacological and genetic inhibition of the 8-OHG–TMEM173 pathway suppresses ferroptosis-mediated pancreatic tumorigenesis. The red line means the overexpression of GPX4 can inhibit the progression of ferroptosis.

**Figure 3 ijms-22-10944-f003:**
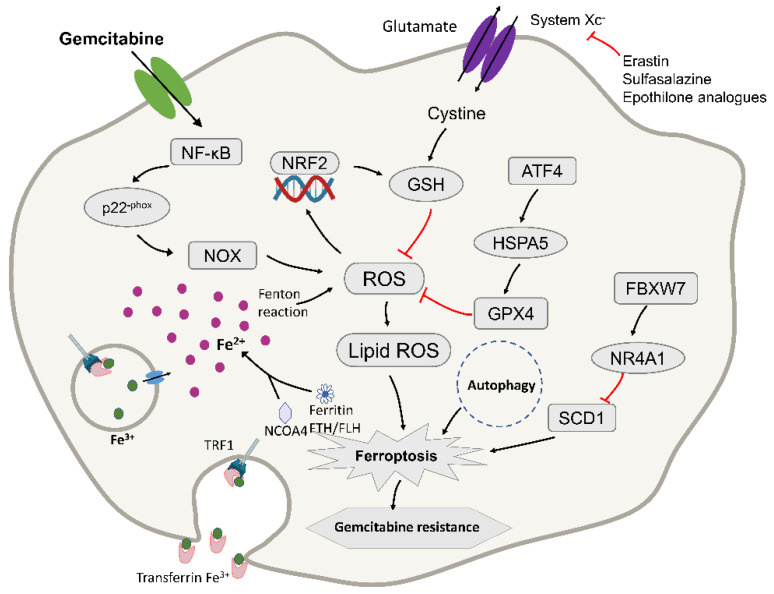
Diagram depicting the molecular targets of ferroptosis and gemcitabine in pancreatic cancer cells. TFR1, transferrin receptor 1; NRF2, nuclear factor erythroid 2-related factor 2; ROS, reactive oxygen species; GPX4, glutathione peroxidase-4; FtH/FtL, ferritin heavy chain/ferritin light chain; NCOA4, nuclear receptor coactivator 4. ATF4, activating transcription factor 4; HSPA5, Heat Shock Protein Family A (Hsp70) Member 5; FBXW7, F-box and WD repeat domain-containing 7; NR4A1, receptor subfamily 4 group A member 1; SCD1, stearoyl-CoA desaturase. ROS accumulation results in the activation of NRF2 and an increase in cellular GSH levels, contributing to intrinsic resistance in PDAC. It may to fight pancreatic cancer cells by inducing ferroptosis. The red lines mean that the expression of upstream molecules can inhibit the progress of downstream molecules.

## Data Availability

Not applicable.

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
