# Peer review of "Ferroptosis: At the Crossroad of Gemcitabine Resistance and Tumorigenesis in Pancreatic Cancer"

_ijms, 2021, doi:10.3390/ijms222010944_

Round 1
Reviewer 1 Report
Yang and colleagues’ Review on Ferroptosis implication in overcoming resistance to gemcitabine is of interest; however, the manuscript is very long and the narrative is not fluid. There is a lot of repetition of facts. The full manuscript should be comprehensibly revised, shortened and restructured.
Specifically, it is not suitable for publication in the current version for the following reasons:
The subtitles, such as in 2.1. and 2.2., though not limited, should better reflect the content below. Furthermore, it is not clear why 2.3. section is an attempt to explore pathways of gemcitabine resistance related to ferroptosis, though it seems MAPK pathway is not one of those related to ferroptosis. In fact, is PI3K/AKT signalling an arm of the MAPK pathway?
I have to disagree that there might have been specifically “doctors” (line 90) that have made progress in immunotherapy, for instance; I would suggest this terminology was avoided. The use of “People” (line 95), “problems” (line 94-5), “scholars” (lines 184, 188) should also be revised.
In 2.3.1 (line 137), 3. (line 289, after PDAC), line 432 (after 2012), etc, a reference is needed.
Iron death definition should be given earlier.
Could the authors better clarify the definition of “external resistance” (line 101).
Figure 1 should be improved; specially the top part, it is not very useful neither truly fits with the text above; microRNA have been included though some were never referenced in the text, while some were in section 2.3.4, but the Figure was not referenced there, and some microRNA were referenced in text (line 233), but not in the Figure; all abbreviations should be disclosed at Figure legends (for all Figures). Regarding Figure 3, is FWB7 (at the image) the same as FBXW7 (in text)?
There are full copied paragraphs, such as in lines 47-58 versus 61-72; other sentences, though not exactly copied, are very repetitive, such as in lines 239-43 versus 246-8 versus 265-7, line 343-4 versus 345-6, or in line 118-120 versus 128-9, but might not limited to these examples.
There is a sense of lack of care in the manuscript, reflected by extensive typographic errors or mispositioned words or pieces of sentences, such as in lines 143, 210, 222, 241, 256, 264, 294, 302, 326, 397, but not limited to these examples. In addition, dots should not be before references.
Early indication of which other patients might receive gemcitabine (line 58) is recommended.
The suitability of references [39,40] in lines 144 should be revised.
It should be clarified whether information from lines 161-3 refers only to breast and colon, or also PDAC.
In line 169, as well as in line 405, it is not clear which report/study the authors are referring to.
The meaning of the sentence in lines 254-5 should be revised. Sentences construction should be revised (ex.: 290-2; 306-7; 326-8), so the message is clearer.
Conclusions should be improved. From line 432 to 441 it seems a repetition of the Introduction. Perspectives were omitted.
Author Response
Please see the attachment。

Reviewer 2 Report
Dear authors,
I have read your paper with interest. It is well written, the conceptualization is original and the overall writing was well-conducted.
My only observation is related to the bibliography. From 117 titles, you have only chosen 5 new resources to cite (4 from 2020 and only 1 from 2021). I have conducted a simple Web of Science search and I have found 1020 article titles by searching ”ferroptosis”, which were published only in 2021 so far. Searching ”ferroptosis AND pancreatic cancer” in 2021, I have still retrieved 29 results. This topic looks like it has become highly popular nowadays, and it would be interesting if the authors discuss in this paper the most freshly published literature in this field. My advice is to take some time to review the 2020-2021 literature on this field, maybe it will bring a real improvement to this paper. Please take only as a personal opinion my advice, maybe you will still find the recent articles as unsuitable for being cited in your paper.
Good luck!
Round 2
Reviewer 1 Report
The authors have in general answered my concerns.
Nevertheless, before publication, I would recommend the revision to my previous question regarding PI3K/AKT pathway being considered an arm of the MAPK pathway, which the authors did not modify. Though both pathways are downstream of RTKs (and PI3K may be activated by RAS), stated like the authors have in the manuscript, it is not accurate. PI3K is downstream of RAS activation, but this is not an arm of the RAF-MEK-ERK pathway (though also activated by RAS), also called MAPK pathway. Please correct before final acceptance.
Author Response
Dear Reviewer:
Thank you for your praise and for your comments concerning our manuscript entitled “Ferroptosis:at the crossroad of gemcitabine resistance and tumorigenesis in pancreatic cancer” (ID: ijms-1399594) again. Those comments are all valuable and very helpful for revising and improving our paper, as well as the important guiding significance to our researches. We have studied comments carefully and have made correction which we hope meet with approval. Revised portion are marked in red in the paper. The main corrections in the paper and the responds to your comments are as following:
- Response to comment: Nevertheless, before publication, I would recommend the revision to my previous question regarding PI3K/AKT pathway being considered an arm of the MAPK pathway, which the authors did not modify. Though both pathways are downstream of RTKs (and PI3K may be activated by RAS), stated like the authors have in the manuscript, it is not accurate. PI3K is downstream of RAS activation, but this is not an arm of the RAF-MEK-ERK pathway (though also activated by RAS), also called MAPK pathway. Please correct before final acceptance.
Q&A:Considering your suggestions, we have revised this part. We have carefully considered your opinions, and after consulting the literature, we revised the article on the issues of MAPK pathway and PI3K/AKT pathway in the article.
We tried our best to improve the manuscript and made some changes in the manuscript. These changes will not influence the content and framework of the paper. And here we did not list the changes but marked in red in revised paper.
We appreciate for your warm work earnestly, and hope that the correction will meet with approval.
Once again, thank you very much for your comments and suggestions.
